# Respiratory Syncytial Virus-Infected Human Mesenchymal Stem Cells Overexpress Toll-like Receptors and Change the Pattern of Distribution of Their Cytoskeleton

**DOI:** 10.3390/v17060763

**Published:** 2025-05-28

**Authors:** César Alexis Rosales Velázquez, Laura Guadalupe Chavéz Gómez, Carlos Arturo Félix Espinosa, Mario Adan Moreno-Eutimio, Juan José Montesinos, Guadalupe R. Fajardo-Orduña, Rocio Tirado Mendoza

**Affiliations:** 1Laboratorio de Biología del Citoesqueleto y Virología, Departamento de Microbiología y Parasitología, Facultad de Medicina, Universidad Nacional Autónoma de México, Circuito Interior sin Número, Coyoacán 04510, Mexico; 312092900@quimica.unam.mx (C.A.R.V.); lauraqfb04@gmail.com (L.G.C.G.); 317213445@quimica.unam.mx (C.A.F.E.); 2Departamento de Biología, Facultad de Química, Universidad Nacional Autónoma de México, Edificio A, Circuito Escolar sin Número, Coyoacán 04510, Mexico; marioadan@quimica.unam.mx; 3Laboratorio de Células Troncales Mesenquimales, Unidad de Investigación de Oncología, Hospital de Oncología, Centro Médico Nacional, IMSS, Cuauhtémoc 06720, Mexico; montesinosster@gmail.com (J.J.M.); guadalupefajardo@hotmail.com (G.R.F.-O.)

**Keywords:** human respiratory syncytial virus, mesenchymal stem cells, viral infection, Toll-like receptor expression, stemness biomarkers, cytoskeleton

## Abstract

Acute respiratory tract infections (ARIs) are one of the major causes of morbimortality in children and adulthood. Furthermore, the respiratory syncytial virus (RSV) is the main pathogen in severe lower respiratory tract infections. In Mexico, RSV is the second cause of ARI, affecting mainly children and seniors. RSV infects the airway epithelium, including mesenchymal stem cells (MSCs). These cells express a variety of surface molecules which may function as viral receptors, i.e., Toll-like receptors (TLRs), but the consequences that viral infection has on their biological activities are poorly understood. The aim of this study is to determinate if RSV infection of MSC modifies the expression of stemness biomarkers, TLRs, and the organization of the cytoskeleton. To study the viral infection of MSCs, we determined the mRNA expression using qRT-PCR of SOX2, NANOG, and POU5F1; vimentin and actin; and TLRs 2, 4, and 6. In addition, we determined the cell surface expression of TLR 2 and 4 using flow cytometry. Our results showed that the infection did not change the mRNA expression of SOX2, NANOG, and POU5F1, but increased the mRNA expression of TLR4 and the cell surface expression. Meanwhile, the mRNA in the actin was unchanged, vimentin decreased, and the infection generated a redistribution of the cytoskeleton.

## 1. Introduction

Acute respiratory tract infections (ARIs) are one of the leading causes of morbidity and mortality among infants and young children worldwide [1,2]. Furthermore, it is important to mention that the human respiratory syncytial virus (RSV) is one of the most important pathogens that cause severe lower respiratory tract infection, such as bronchiolitis, which, in some cases, has fatal consequences [3,4]. In Mexico, during the 48th epidemiological week (24 November 2024), RSV is the first cause of respiratory viral infection (40.9%), with the main affected population being children under 9 years of age and adults over 65 [5]. RSV infects cells along the airway epithelium, including the lung-resident mesenchymal stem cells/progenitor cells (MSCs) which may have a biological impact on the pulmonary microenvironment [6]. Furthermore, previous reports showed the susceptibility and permissiveness of MSCs to RSV infection [7]. These MSCs are positive for CD90, CD105, and CD73 and negative for CD45, CD19, CD14, CD11b, CD34, and DR human leukocyte antigens (HLA-DRs) [8]. In addition, they express the transcription factors of pluripotency or stemness biomarkers POU5F1, SOX2, and NANOG [9] that indicate the undifferentiated states of the cell. These cells express a variety of surface molecules which may function as viral receptors, among them are the Toll-like receptors (TLRs), but the consequences the viral infection has on the biological activities of the MSCs are poorly understood. Likewise, the expressed TLRs in different types of MSCs could contribute to different biological activities such as basal motility, self-renewal, differentiation, and immunomodulation [10]. According to these biological functions, MSCs participate in both the development and the homeostasis of many tissues. It is because of their importance in maintaining tissues that their proliferation and differentiation are highly regulated in these cells. [11,12,13]. Many factors include immune cells, inflammatory mediators, toxins, infectious agents, free radicals, endothelial cells, and epithelial cells, all of which orchestrate changes in the microenvironment of injured tissues that result in the mobilization and differentiation of MSCs [14]. These MSCs can be tissue-resident or be recruited in the injured tissues from stem cell niches. Once the MSC reaches the injured tissue, all the microenvironment factors implicated in the cell damage can stimulate these cells to release growth factors to promote differentiation to other cell lines, such as fibroblasts, endothelial cells, and tissue progenitor cells, which perform tissue regeneration and repair, therefore recovering tissue homeostasis [15,16,17,18].

In the pathogenesis of respiratory viral infections, such as RSV infection, specific cell lines participate in the resolution of immune response and repair of tissue damage. Such is the case of resident lung mesenchymal cells, which also have immunomodulatory capacity. However, the infection of resident lung mesenchymal stem cells caused by RSV triggers the production of soluble mediators of the immune response [12,14,19], which can have a biological impact on the lung microenvironment and, in consequence, they do not participate efficiently in tissue regeneration affecting tissue homeostasis [6]. Bearing in mind all the previously described elements, we propose determining if the RSV infection of the mesenchymal stem cells modifies the expression of molecular receptors such as the TLRs (2, 4, and 6), which could be implicated in the viral entrance and could contribute to the MSC basal motility that might be associated with changes in the expression and/or distribution of the cytoskeleton proteins. In addition, we studied whether RSV infection induces changes in the plasticity properties of MSCs associated with the differentiation capacity evaluated by stem biomarkers such as SOX2, NANOG, and POU5F1.

## 2. Materials and Methods

### 2.1. Cells and Viruses

The 5 batches of placental MSC (MSC-PL) cultures were obtained from 4 donors characterized in the mesenchymal stem cell laboratory in the charge of Dr. Montesinos. Research Committees of the Research Division of the Faculty of Medicine of UNAM (FMED/CI/SPLR/004/2016).

Human type 2 laryngeal epithelial cells (HEp-2; ATCC CCL23,USA; which reported contamination with HeLa cells) [20] were used to multiply viral stocks and for the viral titer test. Human lung epithelial cells type II (A549; ATCC CCL185) were used as the positive control for the qRT-PCR assay to establish the cycle threshold (Ct) value of the target gene (TLRs).

RSV strain Long [subgroup A (RSV-A; ATCC^®^ VR-26™)]. The procedures for propagating viruses and assessing viral infectivity are described in Payment and Trudel (1993) [21].

### 2.2. Infection of MSC-PL

Monolayers of confluent of MSC-PL (pass R3 or R4) were infected with RSV at a multiplicity of infection of 1.0. The infection was performed for 72 h, as previously published [7]. In parallel, the mock-infected MSCs were implemented as experimental controls. The supernatants were collected 72 h after infection and were titrated with TCID50 (infectious dose of tissue culture). TCID50 was calculated according to the Kärber formula [21].

### 2.3. qRT-PCR

Total RNA was isolated from MSC-PL (R3 or R4) using Trizol at 72 h PI. The qRT-PCR assay was performed according to previously standardized conditions in the laboratory. In addition, the sequence of oligonucleotides for the amplification of viral genes was previously published [22]. Amplification assays for stemness biomarkers (POU5F1, NANOG, and SOX2), TLR receptors, and cytoskeleton components (vimentin and actin) were performed according to previously standardized methods and with the following oligonucleotides (Table 1).

The amplification was performed using one-step RT-PCR reagents in accordance with the manufacturer’s instructions (Applied Biosystems). The MyGo Mini (HTF-4FR-NTN) System was used for RT-qPCR and analyzed with the software MyGo v. 3.3.1. The cycle threshold (Ct) value of the target gene was ascertained for TLRs. As a negative control of amplification, we used murine βActin (mActb), and as an internal control, we used hGAPDH.

### 2.4. Staining and Flow Cytometry Analysis

The MSCs (infected for 72 h) at passages 3 and 4 were collected, washed with cold 1X PBS, and resuspended in TruStain FcX™ blocking solution (BioLegend, San Diego, CA, USA) for 10 min at 4 °C to reduce nonspecific antibody binding. Cells were then incubated simultaneously with biotinylated anti-human TLR4 monoclonal antibody (BD Biosciences, San Jose, CA, USA) and Alexa Fluor 647-conjugated anti-human TLR2 monoclonal antibody (BD Biosciences) for 30 min at 4 °C in the dark. After this incubation, cells were washed with 1X PBS and further incubated with APC-Cy7-conjugated streptavidin (BD Biosciences) for 20 min at 4 °C in the dark, in order to detect biotinylated TLR4. Finally, cells were washed, resuspended in 1X PBS, and acquired using an Attune NXT flow cytometer (Thermo Fisher Scientific, Waltham, MA, USA) at the Laboratorio Nacional de Citometría de Flujo (LABNALCIT). Appropriate isotype controls were included to assess signal specificity. Data were analyzed using the FlowJo C 10.10 software (BD Biosciences).

### 2.5. Determination of the Distribution of the Cytoskeleton Proteins in RSV-Infected Mesenchymal Stem Cultures

The immunodetection of filamentous actin was performed by direct labeling with Phalloidin TRIC Labeled (Sigma, Darmstadt, HE, Germany) used at a dilution of 1:100. For the visualization of cell nuclei, the samples were washed with PBS, incubated with 4’,6-Diamidino-2-phenylindole (DAPI; Sigma-Aldrich, Darmstadt, HE, Germany) for 30 min in the dark. Finally, the cell specimens (R3 or R4) were washed and mounted on slides that were covered with the commercial solution Vectashield^®^ (Vector H-1200) at room temperature. All the immunofluorescence-stained images were obtained under a Nikon TS100 microscope (Nikon Co., Melville, NY, USA) using the microscopy program NIS-Elements Viewer 4.2 (Nikon Co., Melville, NY, USA) and they were analyzed on Image J 1.5b software. All the assays were performed in parallel with mock-infected cells.

### 2.6. Statistical Analysis

The data were analyzed and visualized using the statistical software GraphPad Prism version 10.4.1. The analysis of the data was conducted using the unpaired Student’s *t*-test, and the mean values were subsequently subjected to the Mann–Whitney significance test. The statistical significance of all results was determined by *p* ≤ 0.05.

## 3. Results

### 3.1. Characterization of Mesenchymal Stem Cells

The MSC cultures were characterized by Montesinos’s laboratory [23]. Briefly, the first criterion that was taken into consideration was morphology. The MSC-PL shows the typical fibroblastoide morphology, with these cells corresponding to 83.97% of all cells present in culture. The second one is related to phenotypic analysis based on the identification by flow cytometry of the cell-surface molecule characteristics of MSC-PL. The cells tested positive to CD73 (99 ± 1), CD90 (53 ± 20), and CD105 (94 ± 5), likewise to HLA ABC (97 ± 2), and negative to hematopoietic (CD14, CD34, and CD45) and endothelial markers (CD31). They were also negative for CD62L (L-selectin), CD34r, and HLA-DR (Class II). Finally, the differentiation capacity of MSCs derived from MSC-PL was analyzed in culture under conditions that favor adipogenic, osteogenic or chondrogenic differentiation, which were stained with differential dyes to each lineage (Figure 1).

### 3.2. Virus Production in MSC-PL

We performed infection of MSC-PLs with RSV at a moi of 1 in confluent cell cultures (>80%). We observed that syncytia formation begins to occur at 72 h postinfection. Subsequently, the samples were subjected to various analytical procedures, including qRT-PCR, flow cytometry, and immunofluorescence assays. Furthermore, viral titration was conducted on the culture media from the infected MSC-PLs. The viral titter was determined to be(1)$$\frac{10^{-4.578}}{0.025\;\mathrm{mL}}$$

### 3.3. Changes in the mRNA Expression of Stemness Biomarkers (SOX2, NANOG, and POU5F1) by RSV MSC-PL Infection

The previously characterized MSC-PLs proved their capacity to differentiate from the typical lineage; however, we studied the effect the RSV infection has on the expression of MSC-PL stemness biomarkers, and we wondered if the viral infection induces changes over these transcription factors that together regulate the expression of the NANOG. Our results showed that the viral infection at a moi of 1 for 72 h postinfection left the mRNA expression of the three biomarkers SOX2-POU5F1 and in consequence the NANOG as well. The data obtained from the qRT-PCR assay did not demonstrate a statistically significant difference between the RSV-infected MSC-PL cultures versus the mock-infected MSC-PL (Figure 2). These data suggest that RSV infection does not modify the expression of the stemness biomarkers and might cause the MSC-PL culture to differentiate to a particular lineage.

### 3.4. Changes in mRNA and Surface Expression of TLRs upon Infection by RSV

The presence of TLRs in MSCs provided a valuable avenue to investigate the infection of RSVs in these cells. The elucidation of the role of these receptors in the infection process was a critical component of the present study, in which we examined the expression of selected TLRs (2, 4, and 6) in two processes: mRNA transcription and surface expression. Both results were obtained in MSC-PL infected with RSV at a moi of 1. Upon the subsequent linkage of the amplification products of the qRT-PCR assay, a significant difference was observed in TLR4 (*p* = 0.0500) (Figure 3). Conversely, TLR2 and TLR6 exhibited no alterations. In addition, the flow cytometry assay was utilized to examine the expression of TLR2 and TLR4 in MSC-PL infected with RSV at two distinct passages (R3 and R4). The results demonstrated a statistically significant difference in TLR4 expression in both passages R3 and R4 (*p* = 0.011 and *p* = 0.0019, respectively), while TLR2 expression remained unchanged in both passages (Figure 4).

### 3.5. MSC-PL Cytoskeleton Rearrangement as a Result of the RSV Infection

The cytoskeleton has relevant participation in diverse biological properties such as migration, intracellular transport, cell morphology (cellular architecture), tight junctions, cell division, and even differentiation, which implies morphological changes and cell migration. Moreover, the cytoskeleton is implicated during viral infections, mainly in those in which the virus produces syncytium (cell fusion), such as RSV. Therefore, we studied the effect the RSV has on the cytoskeleton of MSC-PL. With this purpose, we demonstrated the RSV infection of MSC-PL using the immunofluorescence assay. After that, we decided to study if the RSV modifies the level of mRNA expression of actin and vimentin. Both cytoskeleton components are involved in cell fusion, hence why the results from qRT-PCR demonstrated that the viral infection induced a decrease in vimentin mRNA expression (Figure 5A) versus mock-infected MSC-PL. This information is agreed upon with other studies which report for the first time the relevance of vimentin in the migration of MSC. Therefore, low levels or the absence of this protein might affect cell migration during tissue regeneration and cell differentiation. Meanwhile, the actin mRNA expression shows no change between infected MSC-PL versus mock-infected MSC-PL (Figure 5B); in accordance with this result, we decided to determine, by immunofluorescence assay, if the RSV infection modified the cytoskeleton distribution. We observed the typical actin stress fiber organization in the mock-infected MSC-PL cultures (Figure 6B); on the other hand, the viral infection modified the typical organization to an agglomerate rearrangement of the actin fibers in the RSV-infected MSC-PL cultures (Figure 6E,F).

## 4. Discussion

Acute respiratory infections (ARIs) are the main cause of morbidity and mortality due to infectious diseases in the world. They are generally self-limiting and clinically range from asymptomatic to severe, the latter causing histopathological damage to the lung. Recently, some studies reported that the HLA-G molecules have a complex immune regulatory role, these molecules may be either compensatory or pathogenetic in different process like viral infections and chronic inflammatory diseases like asthma [24,25]. Severe infections due to respiratory viruses produce in atopic and non-atopic asthmatics a bronchial hyper-responsiveness in the lower airways characterized by persistent airway inflammation and structural remodeling. Moreover, it has been reported that in infants with soluble HLA-G, plasma levels were significantly higher in subjects with persistent wheezing compared with subjects with transient wheezing [25]; thus, HLA-G is a promising biomarker in allergies or other diseases [26]. During the pathogenesis of a respiratory viral infection, several cell lines participate in the recovery of tissue homeostasis. Especially important are lung resident and recruited mesenchymal cells, these cells repair the lung injury tissue and modulate the immune response. The aim of the current study was to establish an in vitro model of respiratory syncytial virus infection of mesenchymal stem cells with the purpose of searching how the viral infection might affect some of the biological properties of these cells like its role in differentiation and TLR expression and migration (cytoskeleton arrangement), among others. In this study, we demonstrated the following: (1) the RSV infection does not modify the capacity of the MSC-PL to differentiate, our results showed no changes in the mRNA expression of the stemness biomarkers (SOX2, POU5F1 and NANOG); (2) the RSV infection modifies both the mRNA expression, such as the expression of TLR4 on the cell surface, while no changes were observed for TLR2 or TLR6 in mRNA or as cell surface molecules; and (3) the RSV-infected MSC-PLs displayed a reorganization of the actin filaments, even though no changes were detected in the mRNA expression.

(1)The stem cells participate at different levels of biological processes; especially relevant are self-renewal and differentiation. Both activities are regulated by transcription factors that act as molecular rheostats [27]. Thus, sensing self-renewal and poise can induce or suppress the gene expression necessary for differentiation [27]. Among them are SOX2 and POU5F1, which work together to drive the expression of their own genes and many other genes, including NANOG [27]. POU5F1/NANOG/SOX2 is an interconnected regulatory circuit, in which NANOG acts as a direct target for POU5F1/SOX2 binding, which maintains the self-renewal and pluripotency of the stem cells [28]. Recent studies described the nuclear reorganization of transcription factors SOX2 and POU5F1 that play a key role in differentiation and the loss of pluripotency of stem cells by modifying the interaction of these molecules with their chromatin targets [29]. The potentiality and the capacity of differentiation are important events to recover the tissue function. With all this information, our interest is focused on determining whether the RSV infection of MSC-PL modifies the mRNA expression of the interconnected regulatory circuit POU5F1/NANOG/SOX2 and in consequence, alters the potentiality and differentiation capacity. However, our data demonstrated that the RSV infection does not modify the mRNA expression of any of the transcription factors; therefore, the MSC-PL maintains their potentiality, their differentiation capacity, and their stemness markers.(2)The Toll-like receptors (TLRs) crucially participate in the modulation of the immune response. These receptors function as one of the several groups of PRRs (Pattern Recognition Receptors), which recognize different PAMPs or MAMPs and DAMPs (pathogen/microbe-associated molecular patterns and death/damage-associated molecular patterns) [30]. These receptors are not only expressed by immune cells but they are also expressed in multiple cell types in addition to immune cells, including stem cells [31]. Likewise, the expressed TLRs in different types of MSCs could contribute to different biological activities such as basal motility, self-renewal, differentiation, and immunomodulation [31], according to the contribution of TLRs in biological processes previously described for the MSCs. First, we confirmed that the MSC-PL expresses TLRs 2, 4, and 6 because, notably, the TLR expression profile varies within different mesenchymal stromal cell populations. Second, we hypothesized that RSV infection of MSC-PL modifies the expression of the mRNA and surface molecule expression of TLRs 2, 4, and 6. Our data show that the viral infection induces an overexpression of both mRNA and surface molecule expression of TLR4. These data are in agreement with previous reports that indicated that RSV increased TLR4 [32,33]. This increment is associated with the fact that the virus uses this molecule as a viral entry receptor. Another study demonstrated changes in the expression of TLR4 in stem cells of the apical papilla, in which the increase in TLR4 was related to decrease in cell proliferation while increasing differentiation [31]. These data are crucial because the authors describe a function other than immune response mediators developed by TLRs. Finally, our results confirmed that RSV infection of MSC-PL does not generate changes in TLR 2 and 6, but suggests that the increment of TLR4 might be associated with intrinsic biological processes of mesenchymal cells, like self-renewal and differentiation.(3)RSV is known to interact with the cytoskeleton in several ways and at various stages throughout its replication cycle. In addition to transcriptional regulation, the cytoskeleton is also likely to be involved in the directional movement of RSV components, and it has a role in virion morphogenesis and cell fusion [34]. Different studies point out the interaction and transport of viral proteins with cytoskeleton filaments; they report the interaction of the viral proteins, Fusion protein (F), Matrix protein (M), and Ribonucleoprotein complex (RNP), with actin filaments to transport virion components to assembly and budding from the infected cells [35]. Moreover, the viral infection induces the upregulation of five genes associated with the cytoskeleton [36]. With the previous information, we decided to study how the RSV infection modifies the expression of the cytoskeleton in the MSC-PL. Our results demonstrate that viral infection does not affect the mRNA expression of actin; however, it was evident that there is a significant change of the morphology of the MSC-PL in which, associated with a redistribution of the actin filaments, the cells lose their fibroblastoid structure and they form a cellular agglomerate. Furthermore, vimentin plays an important role in cell migration, contraction, proliferation, protein synthesis, gene expression, cell apoptosis, and mechanical force transmission, among which participation in cell migration is an important prerequisite for tissue damage repair and inflammation control [37,38]. Contrary to the results with actin, the RSV infection induces a downregulation of the mRNA expression of vimentin. This result suggests that the process of differentiation might be modified, including the rate of MSC growth, according to the study, which states that mice with vimentin gene knockout had slower MSC growth than wild-type mice, and their differentiation status also differed [39].

## 5. Conclusions

The in vitro model proposed in this study suggests that the respiratory syncytial virus infection of the mesenchymal stem cells of the placenta does not alter the cells’ capacity of potentiality, given that the cells conserved the stemness biomarkers and their capacity to differentiate. Viral infection induces the overexpression of TLR4 and changes the organization and distribution of cytoskeleton proteins. These changes by the viral infection may modify some of the biological properties of mesenchymal stem cells and, consequently, they may not be efficient in the processes of regeneration, migration, and proliferation, and possibly could not re-establish tissue homeostasis.

## Figures and Tables

**Figure 1 viruses-17-00763-f001:**
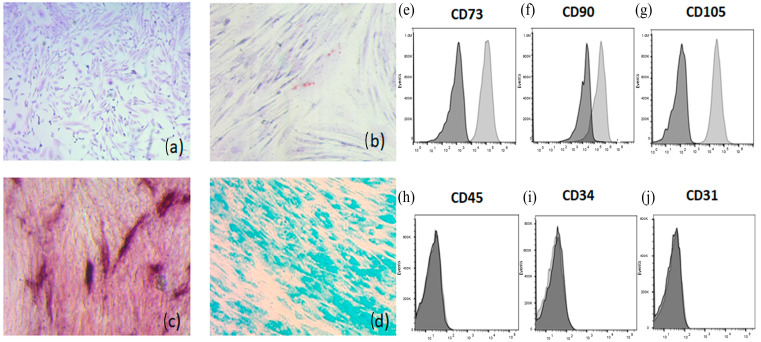
Differentiation of MSC-PL. Briefly the first criterion for the characterization is morphology, (**a**) the MSC-PL monolayer was stained with Toluidine blue to highlight the fibroblastoide shape (magnification 5×), and the second criterion is the differentiation process, (**b**) adipogenic differentiation was indicated by accumulation of neutral lipid vacuoles that were stained with Oil Red O (magnification 20×). (**c**) Osteogenic differentiation was indicated by alkaline phosphatase detection (magnification 20×). (**d**) Chondrogenic differentiation was indicated by chondrogenic matrix colored with Alcian Blue (magnification 20×). The cell-surface molecule characteristics of MSC-PL were analyzed using flow cytometry. Positives markers: (**e**) CD73; (**f**) CD90; and (**g**) CD105. Negative markers: (**h**) CD45; (**i**) CD 34; and (**j**) CD 31.

**Figure 2 viruses-17-00763-f002:**
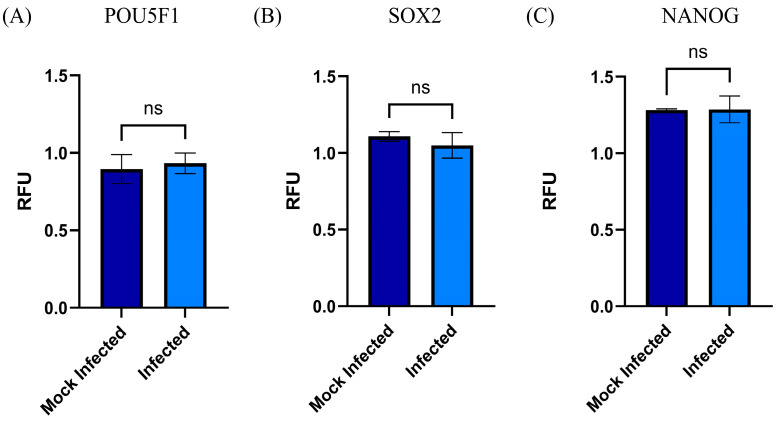
MSC-PL stemness biomarkers. The mRNA expression of the transcription factors (**A**) POU5F1, (**B**) SOX2, and (**C**) NANOG, were analyzed using RT-qPCR with the MyGo Mini System. The analysis was performed in RSV-infected MSC-PLs. Mean values were tested for significance using the Mann–Whitney test. All results were considered significant when *p* ≤ 0.05, ns (not statistically significant). The mock-infected cells were analyzed in parallel.

**Figure 3 viruses-17-00763-f003:**
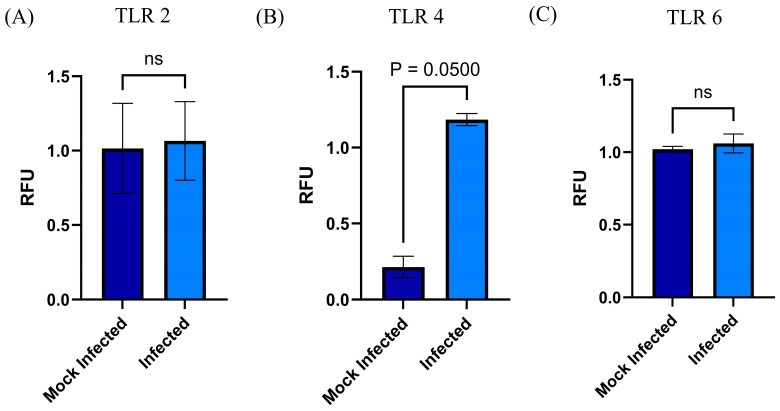
Changes in the expression of TLRs by MSC during RSV infection. The mRNA expressions of (**A**) TLR 2, (**B**) TRL 4, and (**C**) TLR 6 were analyzed by RT-qPCR with the MyGo Mini System. The analysis was performed in RSV-infected MSC-PLs. Mean values were tested for significance using the Mann–Whitney test. All results were considered significant when *p* ≤ 0.05, ns (not statistically significant). The mock-infected cells were analyzed in parallel.

**Figure 4 viruses-17-00763-f004:**
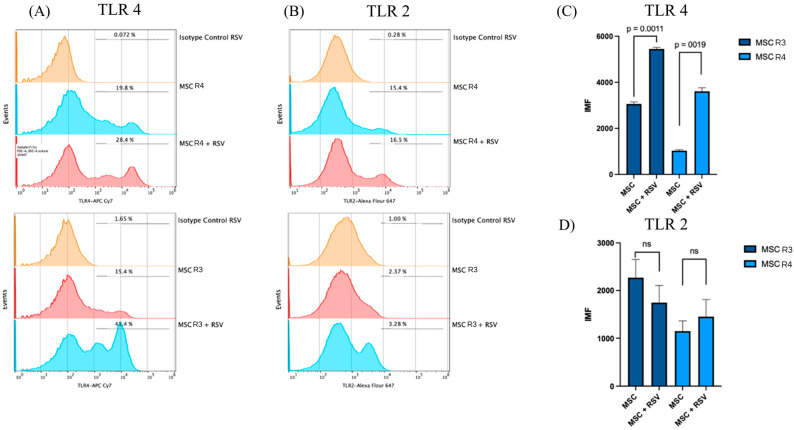
Flow cytometry analysis of TLR4 and TLR2 expressions in MSCs at R3 and R4, with and without RSV infection. Representative histograms show the percentage of TLR4-positive cells (**A**) and TLR2-positive cells (**B**) in MSCs at R3 and R4, under control conditions and after infection with respiratory syncytial virus (RSV). Isotype controls are shown in orange. The bar graphs represent the mean fluorescence intensity (MFI) for TLR4 (**C**) and TLR2 (**D**) in MSC R3 and R4, with and without RSV infection. Statistical analysis was performed using an unpaired *t*-test.

**Figure 5 viruses-17-00763-f005:**
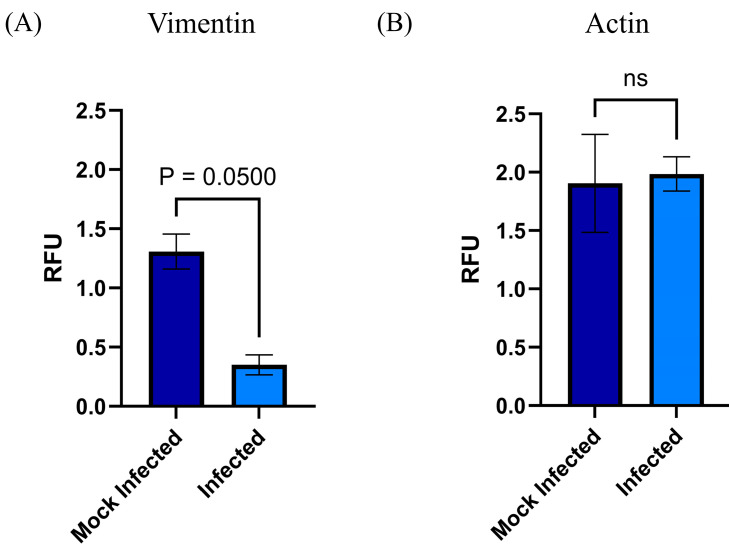
Modification of pattern of mRNA expression of (**A**) vimentin and (**B**) actin in MSC-PL during RSV infection. The mRNA expressions of the actin and vimentin were analyzed by RT-qPCR with the MyGo Mini System. The analysis was performed in RSV-infected MSC-PLs. Mean values were tested for significance using the Mann–Whitney test. All results were considered significant when *p* ≤ 0.05, ns (not statistically significant). The mock-infected cells were analyzed in parallel.

**Figure 6 viruses-17-00763-f006:**
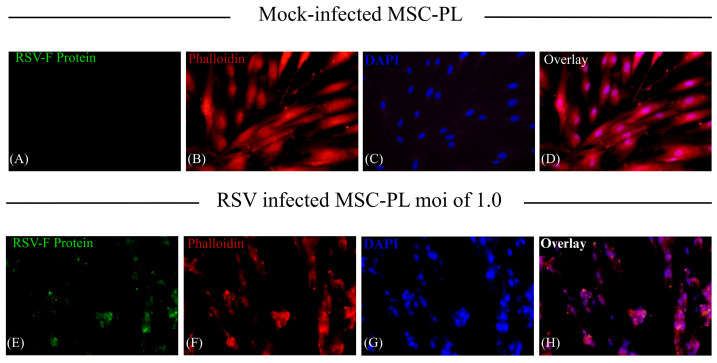
Rearrangement of actin filaments in RSV-infected MSC-PLs. Briefly described, the MSC-PLs were infected with RSV at a moi of 1 over 72 h. The RSV infection and the actin filaments were visualized using immunofluorescence assay. The RSV F viral protein was detected with commercial mABs (antiF0-F1), and the secondary antibody corresponds to anti-mouse-FITC: (**A**) mock-infected and (**E**) RSV-infected MSC-PLs. The detection of filamentous actin was performed by direct labeling with Phalloidin TRIC Labeled: (**B**) mock-infected and (**F**) RSV-infected MSC-PLs. For the visualization of cell nuclei, the samples were incubated with 4’, 6-Diamidino-2-phenylindole (DAPI): (**C**) mock-infected and (**G**) RSV-infected MSC-PLs. Overlay of anti-F (anti-mouse-FITC in green) and actin fibers (Phalloidin TRIC Labeled in red): (**D**) mock-infected and (**H**) RSV-infected MSC-PLs.

**Table 1 viruses-17-00763-t001:** Oligonucleotides.

POU5F1	Forward primer 5′AGTAGTCCCTTCGCAAGCCC3′ Reverse primer 5′CATCGGAGTTGCTCTCCACC3′	386 pb
NANOG	Forward primer 5′GTCTCGTATTTGCTGCATCG3′ Reverse primer 5′ATGCACGTAAGTGCCCTTTT3′	481 pb
SOX2	Forward primer 5′AACCAGCGCATGGACAGTTA3′ Reverse primer 5′GACTTGACCACCGAACCCAT′3′	278 pb
TLR4	Forward primer 5′GAATGCTAAGGTTGCCGCTT3′ Reverse primer 5′TTAGGAACCACCTCCACGC3′	297 pb
TLR2	Forward primer 5′GAGTTCTCCCAGTGTTTGGT3′ Reverse primer 5′TGTTGGAAACTCGAGGCAGA3′	161 pb
TLR6	Forward primer 5′TAGAGGGCTGGCCTGATTCT3′ Reverse primer 5′TGAGAGCCTTCAGCTTGTGG3′	609 pb
F-RSV	Forward primer 5′ATGAACAGTTTAACATTACCAAGTGA3′ Reverse primer 5′CCACGATTTTTATTGGATGCTG 3′	193 pb
Vimentin	Forward primer 5′GGACCAGCTAACCAACGACA3′ Reverse primer 5′GCAGCTCCTGGATTTCCTCT3′	254 pb
Actin	Forward primer 5′GAGCAGTGGCTGAAGGTGAT3′ Reverse primer 5′CAATCCACCTGCCCATGTCT3′	589 pb
mActb	Forward primer 5′GACTTTGTACATTGTTTTG3′ Reverse primer 5′TGCACTTTTATTGGTCTCA3′	382 pb
hGAPDH	Forward primer 5′CCATTCTTCCACCTTTGATGCT3′ Reverse primer 5′TGTTGCTGTAGCCATATTCATTGT3′	87 pb

## Data Availability

The datasets generated and/or analyzed in the current study are available from the corresponding author upon reasonable request.

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
