# Peer review of "Respiratory Syncytial Virus-Infected Human Mesenchymal Stem Cells Overexpress Toll-like Receptors and Change the Pattern of Distribution of Their Cytoskeleton"

_viruses, 2025, doi:10.3390/v17060763_

Round 1
Reviewer 1 Report
Comments and Suggestions for Authors
The paper is interesting and well written. The authors investigated if respiratory syncytial virus (RSV) infection of mesenchymal stem cells (MSC) modifies the expression of stemness biomarkers, TLRs and organization of the cytoskeleton components. The study confirmed that infection unchanged in the mRNA expression of SOX2, NANOG, POU5F1; increases the mRNA of TLR4 and the cell surface of TLR4 expression. I suggest to discuss the role of other molecules as HLA-G in chronic inflammatory immune-mediated diseases (see and add as reference papers by Contini "HLA-G expressing immune cells in immunemediate diseases published in frontiers in immunology in 2020) and viral infections as HIV and HCV (see and add as references papers by Murdaca et a concerning HLA-G and HIV and HCV infections).
Author Response
Comments and Suggestions for Authors
The paper is interesting and well written. The authors investigated whether respiratory syncytial virus (RSV) infection of mesenchymal stem cells (MSC) modifies the expression of stemness biomarkers, TLRs and organization of the cytoskeleton components. The study confirmed that infection unchanged in the mRNA expression of SOX2, NANOG, POU5F1; increases the mRNA of TLR4 and the cell surface of TLR4 expression. I suggest to discuss the role of other molecules as HLA-G in chronic inflammatory immune-mediated diseases (see and add as reference papers by Contini "HLA-G expressing immune cells in immunemediate diseases published in frontiers in immunology in 2020) and viral infections as HIV and HCV (see and add as references papers by Murdaca et a concerning HLA-G and HIV and HCV infections).
Response.
NOTE: The modifications were made in the word version of the manuscript.
We thank the reviewer for the comments and suggestions.
Lines 272-280. The authors have modified the text as suggested by the reviewer to discuss the role of other molecules such as HLA-G. The authors added the following paragraph “Recently, some studies reported that the HLA-G molecules have a complex immune regulatory role, these molecules may be either compensatory or pathogenetic in different process like viral infections and chronic inflammatory diseases like asthma (24,25). Severe respiratory viral infections are a risk factor to developed asthma no atopic and a detonator of bronchial hyperresponsiveness in lower airways in allergic asthma, both are characterized by persistent airway inflammation and structural remodeling. Moreover, it has been reported that in infants with asthma, soluble HLA-G plasma levels were significantly higher in subjects with persistent wheezing compared with subjects with transient wheezing (25); thus, HLA-G is a promising biomarker in allergies or other diseases (26)”.

Reviewer 2 Report
Comments and Suggestions for Authors
In this manuscript, the authors studied whether RSV infection of MSC modifies the expression of stemness biomarkers, TLRs and organization of the cytoskeleton components. Their results showed that the infection did not change the mRNA expression of SOX2, NANOG, POU5F1 whereas it increased the mRNA levels of TLR4 and TLR4 cell surface expression. Moreover,the infection generated a redistribution of the cytoskeleton.
Although the in vitro model proposed in this study suggest that the RSV infection of the MSCs from placenta did not alter their potentiality and their capacity to differentiate, viral infection may modify some of their biological properties and consequently may not be efficient in processes such as regeneration or regulation of tissue homeostasis.
Minor points
1. One of the limitations of the study is the use of MSCs from only one tissue source and one donor. If the authors could add data from additional sources or donors that would increase the value of the results.
2. The authors showed that the viral infection at an MOI of 1 did not change the mRNA expression of the three biomarkers SOX 2, POU5F1 and NANOG at 72 hours postinfection. However, it is possible that the expression of these genes would change at longer times or higher MOIs. Thus, unless the authors perform additional experiments they should discuss this possibility.
3. In lines 184-185, ‘figure 6A’ should be ‘figure 5B’ and in line 189 ‘figure 5’ should be ‘figure 5A’.
4. In line 192, I do not understand the meaning of ‘figure 6 B1’. The same for ‘figure 6 A2 and B2’ in line 194. This is confusing and should be clarified.
5. Figure 4. It would be more clear if the authors add ‘TLR2’ or ‘TLR4’ to the corresponding panels as they did in Figures 2 and 3.
6. In line 139, it should say ‘adipogenic’.
Author Response
Minor points
Comments 1. One of the limitations of the study is the use of MSCs from only one tissue source and one donor. If the authors could add data from additional sources or donors that would increase the value of the results.
Response 1.
We appreciate the reviewer’s observation. Indeed, it is correct that our in vitro model only considers the MSC-PL, however it is important to point out that all the assays were done with MSC cultures obtained from four donors and from these donors 5 batches were obtained with a range of cell numbers from 5x105 to 1x106 cells/ml.
MSC PL (batches) |
R (passages) |
Cell/ml |
03-09 |
R4 |
6.8 x105 |
06-09 |
R4 |
1x106 |
04-05 |
R4 |
8x105 |
05-08 |
R2 |
5x105 |
08-09 |
R2 |
5x105 |
We agree with the comments of the reviewer that suggest the use of different sources or niches of MSC, and we do not rule out that possibility, but all our previous assays and published results were done with MSC-PL and constitute the background information on which this manuscript is based. We have some preliminary results that include assays with umbilical stem cells, these preliminary results suggest some differences in the distribution of some cytoskeleton proteins between MSC-PL versus umbilical, however this consideration must be taken carefully because the umbilical MSC are a cell line.
Lines 86-89. In consideration of the reviewer's comments, the authors suggest modifying the text as follows: “The 5 batches of placental MSC (MSC-PL) cultures were obtained from 4 donors characterized in the mesenchymal stem cell laboratory in charge of Dr. Montesinos. Research Committees of the Research Division of the Faculty of Medicine of UNAM (FMED/CI/SPLR/004/2016)”.
Comments 2. The authors showed that the viral infection at an MOI of 1 did not change the mRNA expression of the three biomarkers SOX 2, POU5F1 and NANOG at 72 hours postinfection. However, it is possible that the expression of these genes would change at longer or higher MOIs. Thus, unless the authors perform additional experiments, they should discuss this possibility.
Response 2.
We appreciate the reviewer’s comment. In a previously published paper of our group (In vitro Pneumovirus and Paramixovirus infection is modulated by the passage of mesenchymal stem cells, by Karla Zarate, Xóchitl Ambriz, Javier R. Ambrosio† and Rocio Tirado. Departamento de Microbiología y Parasitología, Facultad de Medicina, Universidad Nacional Autónoma de México (UNAM), CP 04510, Mexico City, Mexico; https://doi.org/10.54647/cm32819), we established a kinetic of infection (18, 24 48, and 72 hours) at different multiplicities of infection (0.2, 0.5 and 1,0) and we used different number of passages of MSC-PL and different human respiratory viruses. “We found that the viral titer increased with respect to the number of hMSC passages. This coincided with the highest gene expression levels documented at the same passages. As for the hMSC morphological changes, we suggest that these changes were associated with actin modifications. Taken together, viral infections of hMSCs cause altered gene expression and cytoskeleton morphology, with the viral loads ascending as a function of the number of passages”. According to our published data we established the experimental conditions for this manuscript.
NOTE: The modifications were made in the word version of the manuscript.
Comments 3. In lines 184-185, figure 6A’ should be ‘figure 5B’ and in line 189 ‘figure 5’ should be ‘figure 5A’.
Response 3.
We thank the reviewer for the observations.
Line 241. The authors have modified the text to avoid confusion by figure 5A instead of figure 5B.
Line 245. The authors have modified the text to avoid confusion by figure 5B instead of figure 5A.
Line 253. The authors have modified the text to avoid confusions by Vimentin, Actin instead of Actin, Vimentin.
Comments 4. In line 192, I do not understand the meaning of ‘figure 6 B1’. The same for ‘figure 6 A2 and B2’ in line 194. This is confusing and should be clarified.
Response 4.
Lines 247-250. We appreciate the reviewer’s observation, and we agree with the reviewer’s comments, the authors decided to modify the text in the manuscript to avoid confusion like so: “We observed the typical actin stress fibers organization in the mock-infected MSC-PL cultures (Figure 6 B), on the other hand the viral infection modified the typical organization to an agglomerate rearrangement of the actin fibers in the RSV infected MSC-PL cultures (Figure 6 E and F)”.
Comments 5. Figure 4. It would be more clear if the authors add ‘TLR2’ or ‘TLR4’ to the corresponding panels as they did in Figures 2 and 3.
Response 5.
We appreciate the reviewer’s observation, and we agree with the reviewer, the authors have modified Figure 4 as the reviewer suggested.
Comments 5. In line 139, it should say ‘adipogenic’.
Response 5.
Line 162. We thank the reviewer for the observation. Indeed, this mistake, related to adaptogenic, the authors modified to “adipogenic”.

Reviewer 3 Report
Comments and Suggestions for Authors
The manuscript suffers heavily from poor quality of the English language used, resulting in insufficient comprehensibility and lack of focus. To be suitable for mdpi journals, it will be necessary to thoroughly rewrite the manuscript using the help of a native speaker. A correspondingly resubmitted version might be considered for peer review again.
Comments on the Quality of English LanguageThe manuscript suffers heavily from poor quality of the English language used, resulting in insufficient comprehensibility and lack of focus. It will be essential to thoroughly rewrite the manuscript using the help of a native speaker.
Author Response
Comments and Suggestions for Authors
The manuscript suffers heavily from poor quality of the English language used, resulting in insufficient comprehensibility and lack of focus. To be suitable for mdpi journals, it will be necessary to thoroughly rewrite the manuscript using the help of a native speaker. A correspondingly resubmitted version might be considered for peer review again.
Comments on the Quality of English Language
The manuscript suffers heavily from poor quality of the English language used, resulting in insufficient comprehensibility and lack of focus. It will be essential to thoroughly rewrite the manuscript using the help of a native speaker.
Response.
In attendance to the comments and suggestions of the reviewer, the authors requested the revision of the English in order to be considered for peer review again.

Round 2
Reviewer 3 Report
Comments and Suggestions for Authors
The authors have substantially improved the language to an acceptable quality and adequately addressed the points raised by the reviewers. However, there is still a discernibly awkward sentence in a newly added paragraph which cannot be properly understood (l.393-394: "Severe
respiratory viral infections are a risk factor to developed asthma no atopic and a detonator ......")
Additionally, the authors may concisely explain why their established protocol was published in a listed predator journal (ref. 7)
The quality is fine except concerning the point given in my report.
Author Response
Comments 1
Comments and Suggestions for Authors
The authors have substantially improved the language to an acceptable quality and adequately addressed the points raised by the reviewers. However, there is still a discernibly awkward sentence in a newly added paragraph which cannot be properly understood (l.393-394: "Severe
respiratory viral infections are a risk factor to developed asthma no atopic and a detonator ......")
Response 1.
NOTE: The modifications were made in the word version of the manuscript.
We thank the reviewer for the observations.
Line 274-280 (PDF version l.393-394). The authors have modified the text to avoid confusion by “……. Severe infections due to respiratory viruses produce in atopic and non-atopic asthmatics a bronchial hyperresponsiveness in lower airways characterized by persistent airway inflammation and structural remodeling. Moreover, it has been reported that in infants with soluble HLA-G plasma levels were significantly higher in subjects with persistent wheezing compared with subjects with transient wheezing [25]; thus, HLA-G is a promising biomarker in allergies or other diseases [26]”.
Comments 2.
Additionally, the authors may concisely explain why their established protocol was published in a listed predator journal (ref. 7).
Response 2.
We thank the reviewer for the comment.
I really appreciate your comment, and I would like to tell you that before we published our data, I certainly verified that the academic journal was not a predator. I checked the web page https://beallslist.net/. Until I was truly sure about this information, we decided to send our results to publish. Finally, we must inform our accepted and /or published manuscripts to the institution that gave us financial support, so we must be sure that the journals were not predator; and to be accepted as member of the Sistema Nacional de Investigadores, we must report our published manuscripts, including the impact factor, ISSN number and the citations of each paper.
We checked again the Journal of Clinical Medicine, and we do not find as a predator journal.
I am so sorry If I made a mistake or a wrong search about the journal.
